# Convergence Rates of Bayesian Network Policy Gradient for Cooperative Multi-Agent Reinforcement Learning

**Dingyang Chen**
University of South Carolina
dingyang@email.sc.edu

**Zhenyu Zhang**
Amazon
zhenyuzh@amazon.com

**Xiaolong Kuang**
Amazon
xkkuang@amazon.com

**Xinyang Shen**
Amazon
xinyans@amazon.com

**Ozalp Ozer**
Amazon
ozalpo@amazon.com

**Qi Zhang**
University of South Carolina
qz5@cse.sc.edu

## Abstract

Human coordination often benefits from executing actions in a correlated manner, leading to improved cooperation. This concept holds potential for enhancing cooperative multi-agent reinforcement learning (MARL). Despite this, recent advances in MARL predominantly focus on decentralized execution, which favors scalability by avoiding action correlation among agents. A recent study introduced a Bayesian network to incorporate correlations between agents' action selections within their joint policy, demonstrating global convergence to Nash equilibria under a tabular softmax policy parameterization in cooperative Markov games. In this work, we extend these theoretical results by proving the convergence rate of the Bayesian network joint policy with log-barrier regularization.

## 1 Introduction

Cooperative multi-agent reinforcement learning (MARL) enables autonomous agents to collectively maximize their utility, offering promising applications in areas such as traffic control [1], coordination of multi-robot systems [2], and power grid management [3]. However, the joint action space in MARL grows exponentially with the number of agents, posing scalability challenges. To mitigate this, many approaches rely on product policies, where each agent independently selects its action based on its observations. While this simplifies decision-making, it can lead to suboptimal outcomes in cooperative tasks—consider a group of drones performing a synchronized search mission. If each drone acts independently without considering the actions of others, they might overlap in their search areas, wasting resources. In contrast, acting sequentially, where each drone's actions depend on the others, can significantly improve efficiency and coverage. The key research question is how to introduce correlations in multi-agent policies without exacerbating scalability issues. A promising solution is to use a Bayesian network (BN) to capture essential dependencies among agents. This paper provides a theoretical justification on why action dependencies are beneficial.

Recent work has focused on addressing this challenge within Cooperative Markov Games (MGs), a key subclass of Markov games where agents share the same reward function. Policy gradient methods have been shown to converge to Nash policies under product joint policies, with guarantees in tabular settings using both direct [4] and softmax parameterization [5, 6]. However, while product policies offer scalability, they may incur suboptimality for the learned joint policy. To overcome the limitations of product joint policies, a recent advance [7] has introduced the use of Bayesian networks to capture

Workshop on Bayesian Decision-making and Uncertainty, 38th Conference on Neural Information Processing Systems (NeurIPS 2024).

correlations among agents' actions. This approach has demonstrated global convergence to Nash equilibria under a tabular softmax policy parameterization. Nonetheless, challenges remain: these methods lack convergence rate analysis and do not achieve optimality as shown in the single-agent setting, particularly when using dense Bayesian networks that can factorize any joint policy, thereby behaving similarly to the single-agent setting [8]. In this paper, we address these gaps by providing a theoretical justification for the benefits of action dependencies in multi-agent policies, along with a convergence rate analysis using log-barrier regularization. Our results show that it is possible to achieve optimality with a dense Bayesian network, offering a promising direction for improving the performance of cooperative MARL systems.

## 2 Preliminaries

**Cooperative Markov Game.** We consider a cooperative Markov game (MG) $\langle \mathcal{N}, \mathcal{S}, \mathcal{A}, P, r, \mu \rangle$, involving $N$ agents indexed by $i \in \mathcal{N} = \{1, \dots, N\}$. The game consists of a state space $\mathcal{S}$, a joint action space $\mathcal{A} = \mathcal{A}^1 \times \cdots \times \mathcal{A}^N$, a transition function $P : \mathcal{S} \times \mathcal{A} \to \Delta(\mathcal{S})$, a shared team reward function $r : \mathcal{S} \times \mathcal{A} \to \mathbb{R}$, and an initial state distribution $\mu \in \Delta(\mathcal{S})$. Here, $\Delta(\mathcal{X})$ denotes the set of probability distributions over $\mathcal{X}$. We assume full observability, meaning each agent observes the global state $s \in \mathcal{S}$. Under this assumption, a general joint policy $\pi : \mathcal{S} \to \Delta(\mathcal{A})$ maps states to distributions over the joint action space. Given the exponential growth of $\mathcal{A}$ with $N$, the commonly used subclass is the *product policy*, $\pi = (\pi^1, \dots, \pi^N) : \mathcal{S} \to \times_{i \in \mathcal{N}} \Delta(\mathcal{A}^i)$, where the joint policy is factored as a product of local policies $\pi^i : \mathcal{S} \to \Delta(\mathcal{A}^i)$, such that $\pi(a|s) = \prod_{i \in \mathcal{N}} \pi^i(a^i|s)$. The discounted return from time step $t$ is defined as $G_t = \sum_{l=0}^{\infty} \gamma^l r_{t+l}$, where $r_t := r(s_t, a_t)$ is the team reward at time step $t$. The joint policy $\pi$ induces the value function $V_\pi(s_t) = \mathbb{E}_{s_{t+1:\infty}, a_{t:\infty} \sim \pi}[G_t|s_t]$ and the action-value function $Q_\pi(s_t, a_t) = \mathbb{E}_{s_{t+1:\infty}, a_{t+1:\infty} \sim \pi}[G_t|s_t, a_t]$. The cumulative team reward, starting from $s_0 \sim \mu$, is denoted as $V_\pi(\mu) := \mathbb{E}_{s_0 \sim \mu}[V_\pi(s_0)]$. The (unnormalized) *discounted state visitation measure* under policy $\pi$, starting at $s_0 \sim \mu$, is given by:

$$d_\mu^\pi(s) := \mathbb{E}_{s_0 \sim \mu} \left[ \sum_{t=0}^{\infty} \gamma^t \mathrm{Pr}^\pi(s_t = s|s_0) \right],$$

where $\mathrm{Pr}^\pi(s_t = s|s_0)$ is the probability of being in state $s_t = s$ at time $t$ when starting from $s_0$ and following $\pi$. Since all agents share the team reward, the objective in cooperative MARL, as in single-agent RL, is to optimize the joint policy to maximize its value, i.e., $\max_\pi V_\pi(\mu)$. We now define the concepts of $\epsilon$-Nash and $\epsilon$-optimal policies.

**Definition 1** ($\epsilon$-Nash Policy)**.** The *Nash-gap* of a policy $\pi$ is defined as:

$$\mathrm{Nash\text{-}gap}(\pi) := \max_i \left( \max_{\bar{\pi}^i} V_{\bar{\pi}^i, \pi^{-i}}(\mu) - V_\pi(\mu) \right).$$

A product policy $\pi = (\pi^1, \dots, \pi^N)$ is an $\epsilon$-*Nash policy* if $\mathrm{Nash\text{-}gap}(\pi) \leq \epsilon$.

**Definition 2** ($\epsilon$-Optimal Policy)**.** The *optimality-gap* of a policy $\pi$ is defined as:

$$\mathrm{optimality\text{-}gap}(\pi) := \max_{\pi^*} V_{\pi^*}(\mu) - V_\pi(\mu)$$

A policy $\pi$ is an $\epsilon$-*optimal policy* if $\mathrm{optimality\text{-}gap}(\pi) \leq \epsilon$.

## 3 Bayesian Network Joint Policy

In this work, we focus on a class of joint policies introduced in [7] that extends beyond product policies by incorporating action dependencies through a Bayesian network (BN). A BN is represented by a directed acyclic graph (DAG) $\mathcal{G} = (\mathcal{N}, \mathcal{E})$, where $\mathcal{N}$ is the set of vertices (agents), and $\mathcal{E} \subseteq \{(i, j) : i, j \in \mathcal{N}, i \neq j\}$ is the set of directed edges. The parents of an agent $i$ are denoted by $\mathcal{P}^i := \{j : (j, i) \in \mathcal{E}\}$, and their actions are $a^{\mathcal{P}^i} \in \times_{j \in \mathcal{P}^i} \mathcal{A}^j$, as illustrated in Figure 3. Under full observability, a BN (joint) policy $(\pi, \mathcal{G}) = (\pi^1, \dots, \pi^N, \mathcal{G}) : \mathcal{S} \to \Delta(\mathcal{A})$ is defined, where each local policy $\pi^i : \mathcal{S} \times (\times_{j \in \mathcal{P}^i} \mathcal{A}^j) \to \Delta(\mathcal{A}^i)$ depends on both the global state and the actions of its parents. The joint action $a = (a^1, \dots, a^N)$ is then sampled as $\pi(a|s) = \prod_{i \in \mathcal{N}} \pi^i(a^i|s, a^{\mathcal{P}^i})$.

In this formulation, each local policy $\pi^i$ is parameterized using a tabular softmax function:

$$\theta^i = \left\{ \theta^i_{s,a^{\mathcal{P}^i},a^i} \in \mathbb{R} : s \in \mathcal{S}, a^{\mathcal{P}^i} \in \times_{j \in \mathcal{P}^i} \mathcal{A}^j, a^i \in \mathcal{A}^i \right\},$$

leading to the induced softmax local policy:

$$\pi^i_{\theta^i}(a^i|s, a^{\mathcal{P}^i}) \propto \exp(\theta^i_{s,a^{\mathcal{P}^i},a^i}), \qquad (1)$$

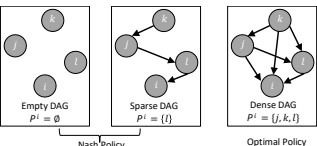

Figure 1: Various BN policies.

with the BN policy parameterized as $\pi_\theta = (\pi^1_{\theta^1}, \ldots, \pi^N_{\theta^N})$.

In Lemma 1, we derive the policy gradient for the BN policy as parameterized in Equation (1), which will be used to establish our convergence results in this section.

We also introduce several shorthand notations. For a subset $\mathcal{M} \subseteq \mathcal{N}$ of agents and its complement $-\mathcal{M}$, a joint action is decomposed as $a = (a^{\mathcal{M}}, a^{-\mathcal{M}})$. The conditional policy for $a^{\mathcal{M}}$ is defined as:

$$\pi^{\mathcal{M}}(a^{\mathcal{M}}|s, a^{-\mathcal{M}}) := \frac{\pi(a^{\mathcal{M}}, a^{-\mathcal{M}}|s)}{\sum_{\bar{a}^{\mathcal{M}}} \pi(\bar{a}^{\mathcal{M}}, a^{-\mathcal{M}}|s)}.$$

The corresponding action-value function is:

$$Q_\pi(s, a^{\mathcal{M}}) := \mathbb{E}_{a^{-\mathcal{M}} \sim \pi^{-\mathcal{M}}(\cdot|s,a^{\mathcal{M}})} \left[ Q_\pi(s, a^{\mathcal{M}}, a^{-\mathcal{M}}) \right].$$

Let $\mathcal{P}^i_+ := \mathcal{P}^i \cup \{i\}$ denote the set of agent $i$ and its parents. We will also use the abbreviations $V_{\pi_\theta}$ and $Q_{\pi_\theta}$ as $V_\theta$ and $Q_\theta$, respectively.

Note that for BN joint policy, the notion of $\epsilon$-Nash policy and $\epsilon$-optimal policy still hold.

## 4 Learning dynamics

### 4.1 Policy gradient dynamics with log-barrier regularization

Building on the method in the single-agent setting [8], we introduce a log-barrier regularized objective, detailed below, to provide finite-time convergence guarantees for policy gradient dynamics:

$$L_\lambda(\theta) := V_\theta(\mu) - \lambda \sum_{i=1}^N \mathbb{E}_{s,a^{\mathcal{P}^i} \sim \mathrm{Unif}_{\mathcal{S} \times \mathcal{A}^{\mathcal{P}^i}}} \left[ \mathrm{KL}(\mathrm{Unif}_{\mathcal{A}^i}, \pi_\theta(\cdot|s, a^{\mathcal{P}^i})) \right]$$

$$= V_\theta(\mu) + \lambda \sum_{i=1}^N \left( \frac{\sum_{s,a^{\mathcal{P}^i},a^i} \log \pi^i_{\theta^i}(a^i|s,a^{\mathcal{P}^i})}{|\mathcal{S}||\mathcal{A}^{\mathcal{P}^i}||\mathcal{A}^i|} + \log|\mathcal{A}^i| \right)$$

where the log barrier regularization, i.e., the KL divergence with respect to the uniform action-selection distribution, is applied to each agent's policy independently.

**Lemma 1** (Tabular softmax BN policy gradient form with log-barrier regularization, proof in Appendix A). For the tabular softmax BN policy parameterized as in Equation (1), we have:

$$\frac{\partial L_\lambda(\theta)}{\partial \theta^i_{s,a^{\mathcal{P}^i},a^i}} = \frac{1}{1-\gamma} d^{\pi_\theta}_\mu(s, a^{\mathcal{P}^i}) \pi^i_{\theta^i}(a^i|s, a^{\mathcal{P}^i}) A^i_\theta(s, a^{\mathcal{P}^i}, a^i) + \frac{\lambda}{|\mathcal{S}||\mathcal{A}^{\mathcal{P}^i}|} \left( \frac{1}{|\mathcal{A}^i|} - \pi^i_{\theta^i}(a^i|s, a^{\mathcal{P}^i}) \right)$$

where $d^{\pi_\theta}_\mu(s, a^{\mathcal{P}^i}) := d^{\pi_\theta}_\mu(s) \sum_{a^{-\mathcal{P}^i}} \pi_\theta(a^{-\mathcal{P}^i}, a^{\mathcal{P}^i}|s)$, $A^i_\theta(s, a^{\mathcal{P}^i}, a^i) := Q_\theta(s, a^{\mathcal{P}^i_+}) - Q_\theta(s, a^{\mathcal{P}^i})$.

The policy gradient form with log-barrier regularizer in Lemma 1 generalizes its counterpart for single-agent policies [8] and for multi-agent product policies [5, 6] under the tabular softmax policy parameterization, which enables us to extend the convergence results to the BN joint policies. Below we state the assumptions used in the convergence results for product policies [5, 6].

**Assumption 1.** For any $\pi$ and any state $s$ of the Markov game, $d^\pi_\mu(s) > 0$.

**Assumption 2** (Reward function is bounded). The reward function $r$ is bounded in the range $[r_{\min}, r_{\max}]$, such that the value function $V$ is bounded as $V_{\min} \leq V_\pi(s) \leq V_{\max} \forall s, \pi$.

We next present our finite-time convergence results for standard policy gradient dynamics with log-barrier regularization in Section 4.2, with proofs in the appendix A.

## 4.2 Convergence rate of Tabular Softmax BN Policy Gradient Dynamics with log-barrier regularization

In Theorem 1, we establish, under the tabular softmax BN policy parameterization, the finite time convergence to a near-Nash policy in a coopertaive MGs of the standard policy gradient dynamics (with log-barrier regularization):

$$\theta_{t+1}^i = \theta_t^i + \eta \nabla_{\theta^i} L_\lambda(\theta_t) \tag{2}$$

where $\eta$ is the fixed stepsize and the update is performed by every agent $i \in \mathcal{N}$.

For each agent $i$, parent actions $a^{\mathcal{P}^i}$, and local action $a^i$, Equation (2) becomes

$$\theta_{s,a^{\mathcal{P}^i},a^i}^{i,t+1} = \theta_{s,a^{\mathcal{P}^i},a^i}^{i,t} + \eta \nabla_{\theta_{s,a^{\mathcal{P}^i},a^i}^i} L_\lambda(\theta_t)(\mu) \tag{3}$$

**Lemma 2.** ((Log barrier regularization's approximate first-order stationary points are near-Nash, proof in Appendix A.2) Suppose $\theta$ is such that $\|\nabla_\theta L_\lambda(\theta)\|_2 \leq \lambda/(2|\mathcal{S}||\mathcal{A}| \max_i |\mathcal{A}^i|)$. Then the BN (joint) policy $\pi_\theta = (\pi_{\theta^1}^1, ..., \pi_{\theta^N}^N, \mathcal{G})$ is a $2\lambda M$-Nash policy where $M := \max_{\pi,\pi'} \left\| \frac{d_\mu^\pi}{d_\mu^{\pi'}} \right\|_\infty$, which is well-defined by Assumption 1.

Lemma 2 extends the results in [8] in the single-agent setting, and the results in [5, 6] in the multi-agent setting with an uncorrelated uncorrelated product policy, stating that, with the log barrier regularization, approximate first-order stationary points in a BN policy are near-Nash. With Lemma 2, we establish the convergence rate as stated in Theorem 1.

**Theorem 1** (Convergence rate of the BN policy gradient with log-barrier regularization, proof in Appendix A). Under Assumptions 1 and 2, suppose every agent $i$ follows the policy gradient dynamics (2), which results in the update dynamics (3) for each each agent $i$, parent actions $a^{\mathcal{P}^i}$, and local action $a^i$, with $\eta \leq \frac{1}{\beta_\lambda}$, then we have $\min_{t<T}$ Nash-gap$_t \leq \epsilon$ whenever

$$T \geq \frac{256 N M^2 |\mathcal{S}|^2 \max_i |\mathcal{A}^i|^2 (r_{\max} - r_{\min})^2}{(1-\gamma)^4 \epsilon^2} + \frac{32 \lambda M |\mathcal{S}| \max_i |\mathcal{A}^i|^2 \sum_i \frac{1}{|\mathcal{A}^{\mathcal{P}^i}|}(r_{\max} - r_{\min})}{(1-\gamma)\epsilon}$$

where $\beta_\lambda = \frac{8N(r_{\max}-r_{\min})}{(1-\gamma)^3} + \sum_i \frac{2\lambda}{|\mathcal{S}||\mathcal{A}^{\mathcal{P}^i}|}$ is an upper bound on the smoothness of $L_\lambda(\theta)$.

The main trick is to view the parent actions $a^{\mathcal{P}^i}$ as part of the state, so that $d_\mu^{\pi_\theta}(s, a^{\mathcal{P}^i})$ becomes the new state visitation measure for the augmented state $(s, a^{\mathcal{P}^i})$. With this transformation, the update dynamics in 1 is similar to the ones for the product policy,i.e., $\mathcal{G} := (\mathcal{N}, \emptyset)$, and thus generalize the results to the BN policy.

With the help of a log-barrier regularizer, we establish convergence to the optimal joint policy with a dense Bayesian network in Theorem 2, an outcome not achieved in [7]. This regularizer ensures that every action has a positive and bounded probability of selection, which is crucial for the sufficient coverage of parent actions and achieving optimality.

**Definition 3** (Fully-correlated BN policy). A BN policy $(\pi, (\mathcal{N}, \mathcal{E}))$ is fully-correlated if $|\mathcal{E}| = N(N-1)/2$, the maximum number of edges in a BN.

**Lemma 3.** ((Fully-correlated BN policy with Log barrier regularization's approximate first-order stationary points are near-optimal, proof in Appendix A.4) Suppose $\theta$ is such that $\|\nabla_\theta L_\lambda(\theta)\|_2 \leq \lambda/(2|\mathcal{S}||\mathcal{A}| \max_i |\mathcal{A}^i|)$. Then the BN (joint) policy $\pi_\theta = (\pi_{\theta^1}^1, ..., \pi_{\theta^N}^N, \mathcal{G})$ is a $2^{N+1} N \lambda M$-optimal policy where $M := \max_{\pi,\pi'} \left\| \frac{d_\mu^\pi}{d_\mu^{\pi'}} \right\|_\infty$, which is well-defined by Assumption 1.

**Theorem 2** (Convergence rate of the fully-correlated BN policy gradient with log-barrier regularization, proof in Appendix A.5). Under Assumptions 1 and 2, suppose every agent $i$ follows the policy gradient dynamics (2), which results in the update dynamics (3) for each each agent $i$, parent actions $a^{\mathcal{P}^i}$, and local action $a^i$, with $\eta \leq \frac{1}{\beta_\lambda}$, then we have $\min_{t<T}$ optimality-gap$_t \leq \epsilon$ whenever

$$T \geq \frac{2^{N+8} N^3 M^2 |S|^2 |\mathcal{A}|^2 \max_i |\mathcal{A}^i|^2 (r_{\max} - r_{\min})^2}{\epsilon^2 (1-\gamma)^4} + \frac{8 N M |S||\mathcal{A}|^2 \max_i |\mathcal{A}^i|^2 (r_{\max} - r_{\min}) \sum_i \frac{1}{|\mathcal{A}^{\mathcal{P}^i}|}}{\epsilon(1-\gamma)}$$

where $\beta_\lambda = \frac{8N(r_{\max}-r_{\min})}{(1-\gamma)^3} + \sum_i \frac{2\lambda}{|\mathcal{S}||\mathcal{A}^{\mathcal{P}^i}|}$ is an upper bound on the smoothness of $L_\lambda(\theta)$.

## 5    Limitations

This work primarily focuses on the theoretical aspects of Bayesian network joint policies without exploring their integration with standard MARL algorithms for practical applications.

## 6    Acknowledgments

Qi Zhang acknowledges funding support from NSF award IIS-2154904 and NSF CAREER award 2237963.

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

# A  Proof of Theorem 1 and Theorem 2

**Lemma 4.** $L_\lambda(\theta)$ is $\beta_\lambda$-smooth, where $\beta_\lambda = \frac{8N(r_{\max}-r_{\min})}{(1-\gamma)^3} + \sum_i \frac{2\lambda}{|\mathcal{S}||\mathcal{A}^{\mathcal{P}^i}|}$

*Proof.* Lemma A.3 in [7] shows that $V_\theta$ is $\frac{8N(r_{\max}-r_{\min})}{(1-\gamma)^3}$-smooth. From the perspective of the augumented state, the Lemma D.4 in [8] becomes that the regularizer for each agent $i$ is $\frac{2\lambda}{|\mathcal{S}||\mathcal{A}^{\mathcal{P}^i}|}$-smooth. Thus, $\beta_\lambda = \frac{8N(r_{\max}-r_{\min})}{(1-\gamma)^3} + \sum_i \frac{2\lambda}{|\mathcal{S}||\mathcal{A}^{\mathcal{P}^i}|}$ is an upper bound on the smoothness of $L_\lambda(\theta)$.  $\square$

## A.1  Proof of lemma 1

*Proof.*

$$\frac{\partial L_\lambda(\theta)}{\partial \theta^i_{s,a^{\mathcal{P}^i},a^i}} = \underbrace{\frac{\partial V_\theta(\mu)}{\partial \theta^i_{s,a^{\mathcal{P}^i},a^i}}}_{(1)} - \underbrace{\frac{\partial \left( \lambda \sum_{i=1}^N \mathbb{E}_{s,a^{\mathcal{P}^i} \sim \text{Unif}_{\mathcal{S}\times\mathcal{A}^{\mathcal{P}^i}}} \left[ \text{KL}(\text{Unif}_{\mathcal{A}^i}, \pi_\theta(\cdot|s,a^{\mathcal{P}^i})) \right] \right)}{\partial \theta^i_{s,a^{\mathcal{P}^i},a^i}}}_{(2)}$$

According to lemma 5.1 in [7],

$$(1) = \frac{1}{1-\gamma} d^{\pi_\theta}_\mu(s, a^{\mathcal{P}^i}) \pi^i_{\theta^i}(a^i|s, a^{\mathcal{P}^i}) A^i_\theta(s, a^{\mathcal{P}^i}, a^i)$$

By the definition of KL Divergence,

$$(2) = \frac{\partial \left( \lambda \sum_{i=1}^N \mathbb{E}_{s,a^{\mathcal{P}^i} \sim \text{Unif}_{\mathcal{S}\times\mathcal{A}^{\mathcal{P}^i}}} \lambda \sum_{i=1}^N \left( \frac{\sum_{s,a^{\mathcal{P}^i},a^i} \log \pi^i_{\theta^i}(a^i|s,a^{\mathcal{P}^i})}{|\mathcal{S}||\mathcal{A}^{\mathcal{P}^i}||\mathcal{A}^i|} + \log |\mathcal{A}^i| \right) \right)}{\partial \theta^i_{s,a^{\mathcal{P}^i},a^i}}$$

$$= - \frac{\lambda}{|\mathcal{S}||\mathcal{A}^{\mathcal{P}^i}|} \left( \frac{1}{|\mathcal{A}^i|} - \pi^i_{\theta^i}(a^i|s, a^{\mathcal{P}^i}) \right)$$

Therefore,

$$\frac{\partial L_\lambda(\theta)}{\partial \theta^i_{s,a^{\mathcal{P}^i},a^i}} = (1) - (2)$$

$$= \frac{1}{1-\gamma} d^{\pi_\theta}_\mu(s, a^{\mathcal{P}^i}) \pi^i_{\theta^i}(a^i|s, a^{\mathcal{P}^i}) A^i_\theta(s, a^{\mathcal{P}^i}, a^i) + \frac{\lambda}{|\mathcal{S}||\mathcal{A}^{\mathcal{P}^i}|} \left( \frac{1}{|\mathcal{A}^i|} - \pi^i_{\theta^i}(a^i|s, a^{\mathcal{P}^i}) \right)$$

$\square$

## A.2  Proof of lemma 2

*Proof.* The proof extends the proof of Theorem 5.2 in [8] by the usage of the multi-agent performance difference lemma (Lemma C.1 in [4]).

Similar to the proof of Theorem 5.2 in [8], we can bound $A^i_\theta(s, a^i) \leq$ for any $(s, a^i)$-pair. It suffices to bound $A^i_\theta(s, a^i)$ for any $(s, a^i)$ where $A^i_\theta(s, a^i) \geq 0$ (else $A^i_\theta(s, a^i) \leq$ is trivially true):

$$\lambda/(2|\mathcal{S}||\mathcal{A}| \max_j |\mathcal{A}^j|) \geq \lambda/(2|\mathcal{S}||\mathcal{A}^{\mathcal{P}^i}||\mathcal{A}^i|) =: \epsilon_{\text{opt}}$$

$$\geq \frac{\partial L_\lambda(\theta)}{\partial \theta^i_{s,a^{\mathcal{P}^i},a^i}} \overset{(i)}{=} \frac{1}{1-\gamma} d^{\pi_\theta}_\mu(s, a^{\mathcal{P}^i}) \pi^i_{\theta^i}(a^i|s, a^{\mathcal{P}^i}) A^i_\theta(s, a^{\mathcal{P}^i}, a^i)$$

$$+ \frac{\lambda}{|\mathcal{S}||\mathcal{A}^{\mathcal{P}^i}|} \left( \frac{1}{|\mathcal{A}^i|} - \pi^i_{\theta^i}(a^i|s, a^{\mathcal{P}^i}) \right) \geq \frac{\lambda}{|\mathcal{S}||\mathcal{A}^{\mathcal{P}^i}|} \left( \frac{1}{|\mathcal{A}^i|} - \pi^i_{\theta^i}(a^i|s, a^{\mathcal{P}^i}) \right)$$

where the last inequality is due to $A_\theta^i(s, a^{\mathcal{P}^i}, a^i) \geq 0$, and by rearranging we get

$$\pi_{\theta^i}^i(a^i|s, a^{\mathcal{P}^i}) \geq 1/2|\mathcal{A}^i| \tag{4}$$

Solving (i) for $A_\theta^i(s, a^{\mathcal{P}^i}, a^i)$, we have

$$A_\theta^i(s, a^{\mathcal{P}^i}, a^i) = \frac{1-\gamma}{d_\mu^{\pi_\theta}(s, a^{\mathcal{P}^i})} \left( \frac{1}{\pi_{\theta^i}^i(a^i|s, a^{\mathcal{P}^i})} \frac{\partial L_\lambda(\theta)}{\partial \theta_{s, a^{\mathcal{P}^i}, a^i}^i} + \frac{\lambda}{|\mathcal{S}||\mathcal{A}^{\mathcal{P}^i}|} \left( 1 - \frac{1}{\pi_{\theta^i}^i(a^i|s, a^{\mathcal{P}^i})|\mathcal{A}^i|} \right) \right)$$

$$\leq \frac{1-\gamma}{d_\mu^{\pi_\theta}(s, a^{\mathcal{P}^i})} \left( 2|\mathcal{A}^i|\epsilon_{\mathrm{opt}} + \frac{\lambda}{|\mathcal{S}||\mathcal{A}^{\mathcal{P}^i}|} \right) \qquad (\pi_{\theta^i}^i(a^i|s, a^{\mathcal{P}^i}) \geq 1/2|\mathcal{A}^i|)$$

$$\leq \frac{2(1-\gamma)\lambda}{d_\mu^{\pi_\theta}(s, a^{\mathcal{P}^i})|\mathcal{S}||\mathcal{A}^{\mathcal{P}^i}|} \qquad (\epsilon_{\mathrm{opt}} = \lambda/(2|\mathcal{S}||\mathcal{A}^{\mathcal{P}^i}||\mathcal{A}^i|)) \tag{5}$$

We are now ready to use the multi-agent performance difference lemma on two BN joint policy with only agent $i$'s parameters changed. For convenience, denote $\sum_{a^{-\mathcal{P}^i}} \pi_\theta(a^{-\mathcal{P}^i}, a^{\mathcal{P}^i}|s)$ as $\overline{\pi}_\theta^{\mathcal{P}^i}(\cdot|s)$ so that $d_\mu^{\pi_\theta}(s, a^{\mathcal{P}^i}) = d_\mu^{\pi_\theta}(s)\overline{\pi}_\theta^{\mathcal{P}^i}(\cdot|s)$.

$\forall i \in \mathcal{N}$, let $\theta'_* = [\theta_*^{-i}, \tilde{\theta}_*^i]$ be the parameters of any joint policy where only agent $i$'s parameters are changed.

By performance difference lemma,

$$V_{\theta'_*} - V_{\theta_*} = \frac{1}{1-\gamma} \mathbb{E}_{\bar{s} \sim d_\mu^{\pi_{\theta'_*}}} \mathbb{E}_{\bar{a} \sim \pi_{\theta'_*}} \left[ A_{\theta_*}(\bar{s}, \bar{a}) \right]$$

$$= \frac{1}{1-\gamma} \mathbb{E}_{\bar{s} \sim d_\mu^{\pi_{\theta'_*}}(\cdot)} \mathbb{E}_{\bar{a}^{\mathcal{P}^i} \sim \overline{\pi}_{\theta'_*}^{\mathcal{P}^i}(\cdot|\bar{s})} \mathbb{E}_{\bar{a}^i \sim \pi_{\tilde{\theta}_*^i}^i(\cdot|\bar{s}, \bar{a}^{\mathcal{P}^i})} \mathbb{E}_{\bar{a}^{-\mathcal{P}_+^i} \sim \pi_{\theta'_*}^{-\mathcal{P}_+^i}(\cdot|\bar{s}, a_+^{\mathcal{P}^i})} \left[ Q_{\theta_*}(\bar{s}, \bar{a}^{\mathcal{P}^i}, \bar{a}^i, \bar{a}^{-\mathcal{P}_+^i}) - V_{\theta_*}(\bar{s}) \right]$$

Since $(\theta'_*)^{-i} = \theta_*^{-i}$ which means $\overline{\pi}_{\theta'_*}^{\mathcal{P}^i}(\cdot|\bar{s}) = \overline{\pi}_{\theta_*}^{\mathcal{P}^i}(\cdot|\bar{s}), \pi_{\theta'_*}^{-\mathcal{P}_+^i}(\cdot|\bar{s}, a_+^{\mathcal{P}^i}) = \pi_{\theta_*}^{-\mathcal{P}_+^i}(\cdot|\bar{s}, a_+^{\mathcal{P}^i})$,

$$= \frac{1}{1-\gamma} \mathbb{E}_{\bar{s} \sim d_\mu^{\pi_{\theta'_*}}(\cdot)} \mathbb{E}_{\bar{a}^{\mathcal{P}^i} \sim \overline{\pi}_{\theta_*}^{\mathcal{P}^i}(\cdot|\bar{s})} \mathbb{E}_{\bar{a}^i \sim \pi_{\tilde{\theta}_*^i}^i(\cdot|\bar{s}, \bar{a}^{\mathcal{P}^i})} \mathbb{E}_{\bar{a}^{-\mathcal{P}_+^i} \sim \pi_{\theta_*}^{-\mathcal{P}_+^i}(\cdot|\bar{s}, a_+^{\mathcal{P}^i})} \left[ Q_{\theta_*}(\bar{s}, \bar{a}^{\mathcal{P}^i}, \bar{a}^i, \bar{a}^{-\mathcal{P}_+^i}) - V_{\theta_*}(\bar{s}) \right]$$

$$= \frac{1}{1-\gamma} \mathbb{E}_{\bar{s} \sim d_\mu^{\pi_{\theta'_*}}(\cdot)} \mathbb{E}_{\bar{a}^{\mathcal{P}^i} \sim \overline{\pi}_{\theta_*}^{\mathcal{P}^i}(\cdot|\bar{s})} \mathbb{E}_{\bar{a}^i \sim \pi_{\tilde{\theta}_*^i}^i(\cdot|\bar{s}, \bar{a}^{\mathcal{P}^i})} \left[ Q_{\theta_*}^i(\bar{s}, \bar{a}^{\mathcal{P}^i}, \bar{a}^i) - V_{\theta_*}(\bar{s}) \right]$$

$$= \frac{1}{1-\gamma} \mathbb{E}_{\bar{s} \sim d_\mu^{\pi_{\theta'_*}}(\cdot)} \mathbb{E}_{\bar{a}^{\mathcal{P}^i} \sim \overline{\pi}_{\theta_*}^{\mathcal{P}^i}(\cdot|\bar{s})} \mathbb{E}_{\bar{a}^i \sim \pi_{\tilde{\theta}_*^i}^i(\cdot|\bar{s}, \bar{a}^{\mathcal{P}^i})} \left[ A_{\theta_*}^i(\bar{s}, \bar{a}^{\mathcal{P}^i}, \bar{a}^i) + Q_{\theta_*}^i(\bar{s}, \bar{a}^{\mathcal{P}^i}) - V_{\theta_*}(\bar{s}) \right]$$

$$= \frac{1}{1-\gamma} \mathbb{E}_{\bar{s} \sim d_\mu^{\pi_{\theta'_*}}(\cdot)} \mathbb{E}_{\bar{a}^{\mathcal{P}^i} \sim \overline{\pi}_{\theta_*}^{\mathcal{P}^i}(\cdot|\bar{s})} \mathbb{E}_{\bar{a}^i \sim \pi_{\tilde{\theta}_*^i}^i(\cdot|\bar{s}, \bar{a}^{\mathcal{P}^i})} A_{\theta_*}^i(\bar{s}, \bar{a}^{\mathcal{P}^i}, \bar{a}^i)$$

$$+ \frac{1}{1-\gamma} \mathbb{E}_{\bar{s} \sim d_\mu^{\pi_{\theta'_*}}(\cdot)} \mathbb{E}_{\bar{a}^{\mathcal{P}^i} \sim \overline{\pi}_{\theta_*}^{\mathcal{P}^i}(\cdot|\bar{s})} \mathbb{E}_{\bar{a}^i \sim \pi_{\tilde{\theta}_*^i}^i(\cdot|\bar{s}, \bar{a}^{\mathcal{P}^i})} \left[ Q_{\theta_*}^i(\bar{s}, \bar{a}^{\mathcal{P}^i}) - V_{\theta_*}(\bar{s}) \right]$$

$$\leq \frac{1}{1-\gamma} \mathbb{E}_{\bar{s} \sim d_\mu^{\pi_{\theta'_*}}(\cdot)} \mathbb{E}_{\bar{a}^{\mathcal{P}^i} \sim \overline{\pi}_{\theta_*}^{\mathcal{P}^i}(\cdot|\bar{s})} \mathbb{E}_{\bar{a}^i \sim \pi_{\tilde{\theta}_*^i}^i(\cdot|\bar{s}, \bar{a}^{\mathcal{P}^i})} \frac{2(1-\lambda)\lambda}{d_\mu^{\pi_\theta}(\bar{s}, \bar{a}^{\mathcal{P}^i})|\mathcal{S}||\mathcal{A}^{\mathcal{P}^i}|}$$

$$+ \frac{1}{1-\gamma} \mathbb{E}_{\bar{s} \sim d_\mu^{\pi_{\theta'_*}}(\cdot)} \mathbb{E}_{\bar{a}^{\mathcal{P}^i} \sim \overline{\pi}_{\theta_*}^{\mathcal{P}^i}(\cdot|\bar{s})} \left[ Q_{\theta_*}^i(\bar{s}, \bar{a}^{\mathcal{P}^i}) - V_{\theta_*}(\bar{s}) \right]$$

$$= \frac{1}{1-\gamma} \mathbb{E}_{\bar{s} \sim d_\mu^{\pi_{\theta'_*}}(\cdot)} \mathbb{E}_{\bar{a}^{\mathcal{P}^i} \sim \overline{\pi}_{\theta_*}^{\mathcal{P}^i}(\cdot|\bar{s})} \frac{2(1-\gamma)\lambda}{d_\mu^{\pi_{\theta_*}}(\bar{s}, \bar{a}^{\mathcal{P}^i})|\mathcal{S}||\mathcal{A}^{\mathcal{P}^i}|}$$

$$= \frac{1}{1-\gamma} \mathbb{E}_{\bar{s} \sim d_\mu^{\pi_{\theta'_*}}(\cdot)} \mathbb{E}_{\bar{a}^{\mathcal{P}^i} \sim \overline{\pi}_{\theta_*}^{\mathcal{P}^i}(\cdot|\bar{s})} \frac{2(1-\gamma)\lambda}{d_\mu^{\pi_{\theta_*}}(s)\overline{\pi}_{\theta_*}^{\mathcal{P}^i}(\cdot|s)|\mathcal{S}||\mathcal{A}^{\mathcal{P}^i}|}$$

$$= \mathbb{E}_{\bar{s} \sim d_\mu^{\pi_{\theta'_*}}(\cdot)} \frac{2\lambda}{d_\mu^{\pi_{\theta_*}}(\bar{s})|\mathcal{S}||\mathcal{A}^{\mathcal{P}^i}|}$$

$$= \sum_{\bar{s}} d_\mu^{\pi_{\theta'_*}}(\bar{s}) \frac{2\lambda}{d_\mu^{\pi_{\theta_*}}(\bar{s})|\mathcal{S}||\mathcal{A}^{\mathcal{P}^i}|}$$

$$\leq \frac{2\lambda}{|\mathcal{A}^{\mathcal{P}^i}|} \max_s \left( \frac{d_\mu^{\pi_{\theta'_*}}(s)}{d_\mu^{\pi_{\theta_*}}(s)} \right) \leq \frac{2\lambda}{|\mathcal{A}^{\mathcal{P}^i}|} M \leq 2\lambda M$$

$\square$

By definition of $\epsilon$-Nash, we know that the BN (joint) policy $\pi_\theta = (\pi_{\theta^1}^1, ..., \pi_{\theta^N}^N, \mathcal{G})$ is a $2\lambda M$-Nash policy.

### A.3 Proof of Theorem 1

Since $L_\lambda(\theta)$ is $\beta_\lambda$-smooth, we have

$$\min_{t \le T} \left\| \nabla_\theta L_\lambda(\theta^{(t)}) \right\|_2^2 \le \frac{2\beta_\lambda(L_\lambda(\theta^*) - L_\lambda(\theta_0))}{T} \le \frac{2\beta_\lambda(V_{\max} - V_{\min})}{T} \le \frac{2\beta_\lambda(r_{\max} - r_{\min})}{T(1-\gamma)},$$

where the last inequality is because we need to choose $T$ large enough such that

$$\sqrt{\frac{2\beta_\lambda(r_{\max} - r_{\min})}{T(1-\gamma)}} \le \lambda/(2|\mathcal{S}| \max_i |\mathcal{A}^i|).$$

Solving the above inequality we obtain $T \ge \frac{8\beta_\lambda|\mathcal{S}|^2 \max_i |\mathcal{A}^i|^2(r_{\max} - r_{\min})}{\lambda^2(1-\gamma)}$. By Lemma 2, we should set $\lambda = \epsilon/2M$ to achieve the specified Nash-gap of $\epsilon$. Plugging in $\lambda = \epsilon/2M$ and $\beta_\lambda := \frac{8N(r_{\max} - r_{\min})}{(1-\gamma)^3} + \sum_i \frac{2\lambda}{|\mathcal{S}||\mathcal{A}^{\mathcal{P}^i}|}$, we have

$$
\begin{aligned}
T &\ge \frac{32M^2|\mathcal{S}|^2 \max_i |\mathcal{A}^i|^2 \beta_\lambda(r_{\max} - r_{\min})}{\epsilon^2(1-\gamma)} \\
&= \frac{256NM^2|\mathcal{S}|^2 \max_i |\mathcal{A}^i|^2(r_{\max} - r_{\min})^2}{(1-\gamma)^4\epsilon^2} + \frac{64\lambda M^2|\mathcal{S}| \max_i |\mathcal{A}^i|^2 \sum_i \frac{1}{|\mathcal{A}^{\mathcal{P}^i}|}(r_{\max} - r_{\min})}{(1-\gamma)\epsilon^2} \\
&= \frac{256NM^2|\mathcal{S}|^2 \max_i |\mathcal{A}^i|^2(r_{\max} - r_{\min})^2}{(1-\gamma)^4\epsilon^2} + \frac{32\lambda M|\mathcal{S}| \max_i |\mathcal{A}^i|^2 \sum_i \frac{1}{|\mathcal{A}^{\mathcal{P}^i}|}(r_{\max} - r_{\min})}{(1-\gamma)\epsilon}
\end{aligned}
$$

### A.4 Proof of lemma 3

*Proof.* By bound on the advantage inequality (5), we know that $\forall s, \mathcal{A}^{\mathcal{P}^i}, a^i$,

$$A_\theta^i(s, a^{\mathcal{P}^i}, a^i) \le \frac{2(1-\gamma)\lambda}{d_\mu^{\pi_\theta}(s, a^{\mathcal{P}^i})|\mathcal{S}||\mathcal{A}^{\mathcal{P}^i}|} = \frac{2(1-\gamma)\lambda}{d_\mu^{\pi_\theta}(s)\pi_\theta(a^{\mathcal{P}^i}|s)|\mathcal{S}||\mathcal{A}^{\mathcal{P}^i}|} \tag{6}$$

By inequality (4), we know

$$\pi_\theta(a^{\mathcal{P}^i}|s) = \sum_{a^{-\mathcal{P}^i}} \pi_\theta(a^{\mathcal{P}^i}, a^{-\mathcal{P}^i}|s) = \sum_{a^{-\mathcal{P}^i}} \prod_j \pi_\theta(a^j|s, a^{\mathcal{P}^j}) \ge \sum_{a^{-\mathcal{P}^i}} \prod_j \frac{1}{2|\mathcal{A}^j|}$$

$$= \frac{1}{2^N} \frac{1}{|\mathcal{A}^{\mathcal{P}^i}|}$$

Plugging in 6, we have

$$A_\theta^i(s, a^{\mathcal{P}^i}, a^i) \le \frac{2^{N+1}(1-\gamma)\lambda}{d_\mu^{\pi_\theta}(s)|\mathcal{S}|} \tag{7}$$

$\square$

Assume without loss of generality that agents $1 \cdots N$ in $\mathcal{G}$ has a topological ordering of $1 \cdots N$ (This means that agent $i \in \mathcal{N}$ is the source of $N - i$ edges and target of $i - 1$ edges).
Note that in this case, $\forall i, a^{\mathcal{P}^i} = [a^{\mathcal{P}_+^{i-1}}]$,

$$
\begin{aligned}
Q^{\pi_\theta, i}(s, a^{\mathcal{P}^i}) &= \mathbb{E}_{\bar{a}^{-\mathcal{P}^i} \sim \pi_\theta(\cdot|s, a^{\mathcal{P}^i})} \left[ Q^{\pi_\theta}(s, a^{\mathcal{P}^i}, \bar{a}^{-\mathcal{P}^i}) \right] \\
&= \mathbb{E}_{\bar{a}^{-\mathcal{P}_+^{i-1}} \sim \pi_\theta(\cdot|s, a^{\mathcal{P}_+^{i-1}})} \left[ Q^{\pi_\theta}(s, a^{\mathcal{P}_+^{i-1}}, \bar{a}^{-\mathcal{P}_+^{i-1}}) \right] = Q^{\pi_\theta, i-1}(s, a^{\mathcal{P}_+^{i-1}}) \tag{8}
\end{aligned}
$$

Follow the reverse topological ordering, $\forall a = [a^{\mathcal{P}^N}, a^N]$,

$$Q^{\pi_{\theta*}}(s, a) = Q^{\pi_{\theta*}}(s, a^{\mathcal{P}^N}, a^N) = Q^{\pi_{\theta*}, N}(s, a^{\mathcal{P}^N}, a^N)$$

By inequality (5),

$$\leq Q^{\pi_{\theta*}, N}(s, a^{\mathcal{P}^N}) + \frac{2^{N+1}(1-\gamma)\lambda}{d_\mu^{\pi_\theta}(s)|\mathcal{S}|}$$

By Equation (8),

$$= Q^{\pi_{\theta*}, N-1}(s, a^{\mathcal{P}^{N-1}}, a^{N-1}) + \frac{2^{N+1}(1-\gamma)\lambda}{d_\mu^{\pi_\theta}(s)|\mathcal{S}|}$$

By inequality (5),

$$\leq Q^{\pi_{\theta*}, N-1}(s, a^{\mathcal{P}^{N-1}}) + 2 * \frac{2^{N+1}(1-\gamma)\lambda}{d_\mu^{\pi_\theta}(s)|\mathcal{S}|}$$

By Equation (8),

$$= Q^{\pi_{\theta*}, N-2}(s, a^{\mathcal{P}^{N-2}}), a^{N-2}) + 2 * \frac{2^{N+1}(1-\gamma)\lambda}{d_\mu^{\pi_\theta}(s)|\mathcal{S}|}$$

By keep doing the same procedure above,

$$\leq Q^{\pi_{\theta*}, 1}(s, a^{\mathcal{P}^1}) + N * \frac{2^{N+1}(1-\gamma)\lambda}{d_\mu^{\pi_\theta}(s)|\mathcal{S}|}$$

Since $a^{\mathcal{P}^1} = \emptyset$,

$$= V^{\pi_{\theta*}}(s) + \frac{2^{N+1}N(1-\gamma)\lambda}{d_\mu^{\pi_\theta}(s)|\mathcal{S}|}$$

By performance difference lemma,

$$V_{\theta_*} - V_\theta = \frac{1}{1-\gamma} \mathbb{E}_{\bar{s} \sim d_\mu^{\pi_{\theta*}}} \mathbb{E}_{\bar{a} \sim \pi_{\theta*}} \left[ A_\theta(\bar{s}, \bar{a}) \right]$$

$$\leq \frac{1}{1-\gamma} \mathbb{E}_{\bar{s} \sim d_\mu^{\pi_{\theta*}}} \mathbb{E}_{\bar{a} \sim \pi_{\theta*}} \left[ \frac{2^{N+1}N(1-\gamma)\lambda}{d_\mu^{\pi_\theta}(s)|\mathcal{S}|} \right]$$

$$= \frac{1}{1-\gamma} \sum_{\bar{s}} d_\mu^{\pi_{\theta*}}(\bar{s}) \left[ \frac{2^{N+1}N(1-\gamma)\lambda}{d_\mu^{\pi_\theta}(s)|\mathcal{S}|} \right]$$

$$\leq \frac{1}{1-\gamma} \sum_{\bar{s}} M \left[ \frac{2^{N+1}N(1-\gamma)\lambda}{|\mathcal{S}|} \right]$$

$$= 2^{N+1}N\lambda M$$

Then, since $\forall s, a, Q^{\pi_{\theta*}}(s, a) \leq V^{\pi_{\theta*}}(s)$, we know that $(\pi_{\theta_*^1}^1, \cdots, \pi_{\theta_*^N}^N, \mathcal{G})$ is an $2^{N+1}N\lambda M$-optimal policy.

## A.5 Proof of Theorem 2

Since $L_\lambda(\theta)$ is $\beta_\lambda$-smooth, we have

$$\min_{t \leq T} \left\| \nabla_\theta L_\lambda(\theta^{(t)}) \right\|_2^2 \leq \frac{2\beta_\lambda(L_\lambda(\theta^*) - L_\lambda(\theta_0))}{T} \leq \frac{2\beta_\lambda(V_{\max} - V_{\min})}{T} \leq \frac{2\beta_\lambda(r_{\max} - r_{\min})}{T(1-\gamma)},$$

where the last inequality is because we need to choose $T$ large enough such that

$$\sqrt{\frac{2\beta_\lambda(r_{\max} - r_{\min})}{T(1-\gamma)}} \leq \lambda/(2|\mathcal{S}||\mathcal{A}| \max_i |\mathcal{A}^i|).$$

Solving the above inequality we obtain $T \geq \frac{8\beta_\lambda |\mathcal{S}|^2 |\mathcal{A}|^2 \max_i |\mathcal{A}^i|^2 (r_{\max} - r_{\min})}{\lambda^2 (1-\gamma)}$. By Lemma 3, we should set $\lambda = \frac{\epsilon}{2^{N+1} NM}$ to achieve the specified optimality-gap of $\epsilon$. Plugging in $\lambda = \frac{\epsilon}{2^{N+1} NM}$ and $\beta_\lambda := \frac{8N(r_{\max} - r_{\min})}{(1-\gamma)^3} + \sum_i \frac{2\lambda}{|\mathcal{S}||\mathcal{A}^{\mathcal{P}^i}|}$, we have

$$
T \geq \frac{2^{N+5} N^2 M^2 |S|^2 |\mathcal{A}|^2 \max_i |\mathcal{A}^i|^2 \beta_\lambda (r_{\max} - r_{\min})}{\epsilon^2 (1-\gamma)}
$$

$$
= \frac{2^{N+8} N^3 M^2 |S|^2 |\mathcal{A}|^2 \max_i |\mathcal{A}^i|^2 (r_{\max} - r_{\min})^2}{\epsilon^2 (1-\gamma)^4} + \frac{2^{N+5} N^2 M^2 |S|^2 |\mathcal{A}|^2 \max_i |\mathcal{A}^i|^2 (r_{\max} - r_{\min}) \sum_i \frac{2\lambda}{|S||\mathcal{A}^{\mathcal{P}^i}|}}{\epsilon^2 (1-\gamma)}
$$

$$
= \frac{2^{N+8} N^3 M^2 |S|^2 |\mathcal{A}|^2 \max_i |\mathcal{A}^i|^2 (r_{\max} - r_{\min})^2}{\epsilon^2 (1-\gamma)^4} + \frac{8NM|S||\mathcal{A}|^2 \max_i |\mathcal{A}^i|^2 (r_{\max} - r_{\min}) \sum_i \frac{1}{|\mathcal{A}^{\mathcal{P}^i}|}}{\epsilon(1-\gamma)}
$$

