# OpenReview forum: "Convergence Rates of Bayesian Network Policy Gradient for Cooperative Multi-Agent Reinforcement Learning"
_NeurIPS.cc/2024/Workshop/BDU — NeurIPS BDU Workshop 2024 Poster_

### Official Review · Reviewer_Fi6p · 2024-09-19
**Good paper, extending previous theoretical results**

**Rating:** 8
**Confidence:** 3

**Review:**

## Summary:

This paper focuses on a setting using Bayesian network to inaugurate correlations between agents’ action selections in their
Joint policy. Previous work has demonstrated a global convergence to Nash equilibria under a tabular softmax policy parameterization under this setting. This paper further gives an analysis of the convergence rate under this setting.

## Strengths:
   - Clarity : The writing is clear and makes it easy for reviewers unfamiliar with the previous work to follow the author's discussion.

   - Quality: The contribution of the paper is clear and important, filling the gaps in previous work.

   - Originality: To the best of my knowledge, this work is novel.

   - Significance:  This paper is important and valuable for the setting the author cares about. But I am concerned about whether this paper can provide insights to the broader MARL community.

## Questions:

- Can learnable DAGs be combined with sequentially updated MARL algorithms, such as HAPPO?
- If they can be combined, can the theoretical properties still be maintained?

---

### Official Review · Reviewer_peaL · 2024-09-21
**This work introduces non-asymptotic convergence results for a policy gradient algorithm for cooperative multi-agent reinforcement learning.**

**Rating:** 7
**Confidence:** 2

**Review:**

This work extends the results from [1], which dealt with learning policies for cooperative Markov games. Instead of a product factorization for the joint policy, the joint policy is modeled as a Bayesian network. The problem setting and motivation are well-explained. While [1] provided an asymptotic convergence result for policy gradient in this setting, in this work the authors provide non-asymptotic convergence results.

The one weakness I wish to point out is that the authors do not provide a discussion on the bounds that they obtain (on lines 122 and 141). The bounds are quite complex, and some insight into how individual variables affect the bound and practical recommendations based on it, for instance, could both enhance readability and make a stronger case for the results of the paper.

This paper also lacks empirical evaluation, which the authors point out as a limitation. That being said, theoretical aspects of learning algorithms, particularly non-asymptotic bounds, are valuable and can be used to guide implementations. I feel that the theorems and their proofs demonstrate sufficient originality for acceptance.

---

## Minor comments
- On line 59, the quantity $V\_{\bar{\pi}^i, \pi^{-i}}$ is not defined.
- I found the notation quite cumbersome, for example, $\nabla_{\theta^i_{s, a^{\mathcal{P}^i}, a^i}}$ on line 110. Using more compact notation could make the paper much easier to read.
---

## References
[1] Dingyang Chen and Qi Zhang. Context-aware Bayesian network actor-critic methods for cooperative multi-agent reinforcement learning. In International Conference on Machine Learning. 2023.

---

### Decision · Program_Chairs · 2024-10-09

Accept (Poster)